# End-to-end Adversarial Sample Generation for Data Augmentation

**Tianyuan Liu** and **Yuqing Sun**[*]
Shandong University
zodiacg@foxmail.com sun_yuqing@sdu.edu.cn

## Abstract

Adversarial samples pose a significant challenge to neural inference models. In this paper, we propose a novel enhancing approach $\mathbf{A}^3$ for the robustness of the neural NLP models, which combines the adversarial training and data augmentation. We propose an adversarial sample generator that consists of a conditioned paraphrasing model and a condition generator. The latter aims to generate conditions which guides the paraphrasing model to generate adversarial samples. A pretrained discriminator is introduced to help the adversarial sample generator adapt to the data characteristics for different tasks. We adopt a weighted loss to incorporate the generated adversarial samples with the original samples for augmented training. Compared to existing methods, our approach is much efficient since the generation process is independent to the target model and the generated samples are reusable for different models. Experimental results on several tasks show that our approach improves the overall performance of the trained model. Specially, the enhanced model is robust for various attacking techniques.

## 1 Introduction

Adversarial samples refer to the inputs that intentionally designed to fool deep learning models (Morris et al., 2020a), which are also called the attack samples (Szegedy et al., 2013). The methods for generating attack samples have deceived many natural language processing (NLP) models, such as text classification (Ebrahimi et al., 2017; Ren et al., 2019), machine translation (Cheng et al., 2020) and etc (Goyal et al., 2022). Along with the application of neural models in various areas, the adversarial samples have drawn much attention.

Generally, the adversarial samples are generated by perturbations on the original sample. To enhance the robustness of NLP neural methods, the

---

*Corresponding Author

adversarial training approach uses the the adversarial samples together with the original samples for training model (Goodfellow et al., 2014b). Miyato et al. (Miyato et al., 2016) proposed the first NLP oriented adversarial training method by applying noise on the word representations. But the perturbed result is in the latent space and does not map to a meaningful sentence. Then many works turned to generating adversarial samples as a means of adversarial training. The often adopted method is the lexical substitution that is formulated as a combinatorial optimization problem and solved by heuristic search algorithms (Yoo and Qi, 2021). Since it requires much interaction with the target model for identifying the important words and selecting the appropriate replacement words, it is time-consuming. Due to the lack of supervised training data, so far there is not applicable end-to-end mode for adversarial generation.

Another kind of approaches to improve model robustness is data augmentation, which aims to generate the similar samples to the original data for increasing the amount of training data. The traditional methods on data augmentation include back-translation (Sennrich et al., 2015), rule-based modification (Wei and Zou, 2019), paraphrasing (Kumar et al., 2019) and other techniques (Li et al., 2022). These methods are based on the distribution of training data, which are model agnostic. However, with these data, the models are still vulnerable to adversarial samples as the attacking techniques are designed for finding the weakness of the target model (Yoo and Qi, 2021).

To solve the above challenges, in this paper, we designed an adversarial sample generation method for data augmentation $\mathbf{A}^3$. We propose an end-to-end adversarial sample generation model. Given an original sample, a condition generator is designed to output a condition which guides a conditioned paraphrasing model to generate a potential adversarial sample. By pretraining an adversarial dis-

criminator to be utilized as the training signal, it can learn the task-specific data characteristics for finding the potential vulnerabilities. Since the generation process doesn't require interaction with the target model, it is much efficient compared with the current search based generation methods. Besides, the generated samples can be reused for different models rather than for a specific model as the traditional way. Then we propose a weighted loss for incorporating the generated adversarial samples with the original data for the target task. We evaluate our method on several NLP tasks and compare it to both the data augmentation and adversarial training methods. Experimental results demonstrate that our approach not only improves the overall performance of the trained model but also enhances its robustness against multiple attacking techniques. Moreover, the computational time is significantly reduced compared to existing adversarial sample generation methods.

## 2 Related works

### 2.1 Adversarial Sample Generation

In the NLP area, the disturbance on the original sample is often adopted in adversarial sample generation due to the discrete character of text, such as word replacement. For example, TextBugger (Li et al., 2019) generates the adversarial samples by randomly replacing the words in a sample text for gathering the candidate samples that may flip the model prediction. Then it chooses the reasonable adversarial samples with the similar semantics with the original text. PWWS (Ren et al., 2019) uses synonym for word substitution with the help of WordNet. With the development of pretrained language models (PLM), it help various NLP tasks to gain improvement on the performance. Jin et al. proposed the TextFooler (Jin et al., 2020) method for evaluating the robustness of BERT. Meanwhile, several works utilizes PLM to generate the candidate adversarial texts with similar semantics to original ones (Li et al., 2020, 2021). Recently, a survey on word substitution based adversarial sample generation (Chiang and Lee, 2022) shows that despite the effectiveness of such attacks, the generated samples have flaws on fluency and syntactical correctness and can be identified easily.

### 2.2 Adversarial Training

Based on the proximity assumption, researchers have explored methods to enhance the robustness

and effectiveness of trained models by introducing disturbances to the original training samples, commonly known as adversarial training (Goodfellow et al., 2014b). Miyato et al. (Miyato et al., 2016) first proposed a NLP-oriented method that adds noise to the word embeddings of the input sample for adversarial training in text classification task. Similar approaches have been applied to adversarial training for pretrained language models (Liu et al., 2020) and other NLP tasks (Li and Qiu, 2021), showing improvements in model performance. Despite their effectiveness, disturbance in the continuous spaces lack clear meaning in actual language, making them less interpretable.

Recent works in adversarial training for NLP tasks consider modifications at the lexical level to make the perturbed samples more consistent with real samples. Early works use synonym substitution to explore proximate samples in the sample space. Huang et al. (Huang et al., 2019) and Dong et al. (Dong et al., 2021) used different sample space exploration methods to find samples that effectively enhance model robustness in adversarial training. The SSMBA method proposed by Ng et al. (Ng et al., 2020) generates perturbed samples by using pretrained language models to complete corrupted texts. Recently, researchers consider using adversarial sample generation methods for adversarial training. Previous work (Carlini et al., 2019) has shown that using samples generated by a specific adversarial sample generation method for augmented training generally only improves the robustness against that method. Yoo et al. designed the A2T method (Yoo and Qi, 2021), which introduced a fast lexical substitution adversarial sample generation method into the model training process, achieving improved robustness against multiple model attacking methods. Current adversarial training methods for NLP models are mostly based on constrained searching in the sample space, and lack the ability to self learn and adapt to models or datasets. Moreover, the lexical substitution based generation methods have potential problems regarding semantic soundness and syntactic correctness. (Chiang and Lee, 2022)

## 3 The End-to-end Adversarial Sample Based Data Augmentation

### 3.1 Framework

In this paper, we design the end-to-end adversarial sample generation method for data augmentation.

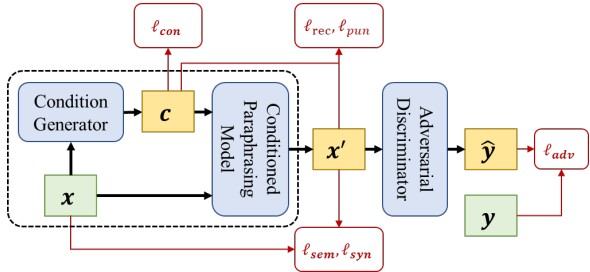

Figure 1: The framework for the end-to-end adversarial sample based data augmentation

For a given sample $\mathbf{x}$, the condition generator gives the condition $\mathbf{c}$, based on which the paraphrasing model generates $\mathbf{x}'$ that is a potential adversarial sample. The whole architecture consists of three parts, as shown in Fig.1. The first *condition generator* is designed to generate the appropriate conditions for guiding the generated paraphrase being an adversarial sample. The second is the pretrained *conditioned paraphrasing model* that generates the paraphrases satisfying the given condition.

Since there are no supervised data for training the condition generator, we propose an *adversarial discriminator* to provide the proper training signals, similar to the structure of a generative adversarial network (GAN) (Goodfellow et al., 2014a). The third part is the adversarial discriminator that can be a classifier or regression model for the target task, which checks the flip of predication.

Then for the target task, the generated adversarial samples are incorporated with the original data for data augmentation, which are used to train a model for this task.

## 3.2   Condition Generator

The condition generator takes an original sample as input and aims to generate a condition $\mathbf{c}$ for guiding the paraphrasing model to generate an adversarial sample. Let $G$ denote the condition generator.

$$\mathbf{c} = G(\mathbf{x}) \qquad (1)$$

There are many candidate forms of condition $\mathbf{c}$, such as the ratio of changed words, the sentence structure or the sentiment. Following the normal way of generating adversarial samples, we adopt the lexical substitution indicator as the condition, which is in the form of binary sequence corresponding to the words in $\mathbf{x}$, denoted by $\mathbf{c} \in \{0,1\}^{|\mathbf{x}|}$. $c_i = 1$ indicates $x_i$ should be changed in the paraphrase $\mathbf{x}'$, $c_i = 0$ for otherwise.

In this paper we adopt a multi-layer bidirectional recurrent neural network as the basic network. The output hidden vector at each step is projected to a scalar. The output of the condition generator is fed into the conditioned paraphrasing model. Since the condition $\mathbf{c}$ is defined as a sequence of binary value, while conventional activation function such as sigmoid outputs a contiguous value between 0 and 1. To maintain a reasonable form of $\mathbf{c}$ and achieve a stable training, we adopt the straight-through estimator (Bengio et al., 2013) method as the activation function, such that the output of the condition generator will be binary.

$$\mathbf{c} = STE(G(\mathbf{x})) \qquad (2)$$

## 3.3   Pretraining the Conditioned Paraphrasing Model

The general purpose of paraphrasing is to generate a sample with the same meaning but different expression with the given text. Here we introduce the conditioned paraphrasing model for guiding the paraphrase. For a given text $\mathbf{x}$ and a condition $\mathbf{c}$, it generates a paraphrase $\mathbf{x}'$ with the similar semantics to $\mathbf{x}$ and satisfying $\mathbf{c}$, formally:

$$\mathbf{x}' = F(\mathbf{x}, \mathbf{c}) \qquad (3)$$

We adopt a multi-layer bidirectional recurrent neural network for the conditioned paraphrasing model. The pretraining data are automatically generated, where the input is a pair of a text $\mathbf{x}$ and the randomly selected positions as the condition $\mathbf{c}$. As the paraphrase $\mathbf{x}'$, there are many candidate methods for choosing the replacement words, such as the synonym vocabulary or the masked language model. To ensure the semantic similarity and the syntactical correctness, we adopt the word composable knowledge (Liu et al., 2023) for choosing the words to form a new text $\mathbf{x}'$ as the supervised target.

Since only a small proportion of the text are changed, the simple negative log-likelihood loss would lead the model to just copy the input. Thus, we apply a larger weight $\omega_p > 0$ for the replacement positions $c_i = 1$:

$$\ell_P = -\sum_{i=1}^{|\mathbf{x}|} (1 + c_i\omega_p) \log p(\hat{x}'_i = x'_i | \mathbf{x}, \mathbf{c}) \quad (4)$$

where $\hat{x}'_i$ is the output of the model and $x'_i$ is the supervised paraphrase.

## 3.4 Adversarial Discriminator

To train the end-to-end adversarial sample generation model, we pretrain an adversarial discriminator to provide training signals, denote by $D$. It is a trained classifier or a regression model for the target task.

$$y = D(\mathbf{x}) \tag{5}$$

Theoretically the paraphrasing model should generate the semantically similar text with the original one. If the prediction $\hat{y}'$ by $D$ for the generated sample $x'$ is flipped compared to the ground truth $y$, $x'$ is a potentially adversarial sample.

## 3.5 End-to-End Adversarial Sample Generation Training

The core of the adversarial sample generation model is to generate the appropriate condition for paraphrasing. Since the discriminator can be seen as a white-box victim model, we can pass the gradient through the neural network structure by utilizing the tricks like Gumbel softmax (Jang et al., 2017). This allows an end-to-end training mode for the whole adversarial sample generation method.

During the training, the condition generator, the conditioned paraphrasing model and the adversarial discriminator are connected in a pipeline way. For a given sample $\mathbf{x}$, the pipeline outputs a prediction $\hat{y}$ for a perturbed sample. The conditioned paraphrasing model $F(\mathbf{x}, \mathbf{c})$ outputs a sequence of word probabilities. We adopt Gumbel softmax to convert the probabilities to word embeddings that can be directly used by $D$.

$$\begin{aligned} \mathbf{c} &= G(\mathbf{x}) \\ \mathbf{x}' &= F(\mathbf{x}, \mathbf{c}) \\ \hat{y} &= D(\mathbf{x}') \end{aligned} \tag{6}$$

During training, the parameters of the condition generator and the conditioned paraphrasing model are tuned while the adversarial discriminator remains frozen. Training the condition paraphrasing model allow the model be trained adaptively to generate words more suitable for the target task. To train the adversarial sample generation model, we used the following losses.

**Adversarial loss** is the main goal of the model, namely to generate a sample that can make the target model flip its prediction. Since previous works on model distilling show the effectiveness and simplicity of the MSE loss function (Hinton et al., 2015), thus it is adopted in our work. More

advanced distances between probabilities such as KL-distance can be an alternative.

$$\ell_{adv} = 1 - |D(\mathbf{x}') - D(\mathbf{x})|^2 \tag{7}$$

**Perturbation ratio loss** To maintain the similar semantics between the perturbed sample and the original sample, the ratio of perturbed words in text are constrained, namely the number of $c_i = 1$ cases in the generated condition $\mathbf{c}$. Let $\rho$ denote the expected ratio, $L$ is the length of sample, the perturbation ratio loss is calculated as follows.

$$\ell_{con} = |\rho - \|\mathbf{c}\|_1 / L|^2 \tag{8}$$

**Conditioned paraphrasing losses** The conditioned paraphrasing model is fine-tuned for different expectation by a few loss functions. The first is the reconstruction loss for the words not perturbed in $\mathbf{c}$, i.e. on the positions $c_i = 0$, denoted by $\mathbf{c}(0)$. It is a negative likelihood loss for the original words such that the paraphrasing model is trained to reconstruct the original input.

$$\ell_{rec} = - \sum_{i \in \mathbf{c}(0)} \log p(\mathbf{x}'_i = \mathbf{x}_i | \mathbf{x}, \mathbf{c}) \tag{9}$$

For the perturbed positions, the probability of the original word should be lower than any other words. Inspired by the loss function for the negative samples in the noise contrastive estimation (Gutmann and Hyvärinen, 2010), we design the loss of punishing repeating the original word. Let $\mathbf{c}(1)$ denote the positions of the 1s in the condition $\mathbf{c}$.

$$\ell_{pun} = - \sum_{i \in \mathbf{c}(1)} \sigma(-\log p(\mathbf{x}'_i = \mathbf{x}_i | \mathbf{x}, \mathbf{c})) \tag{10}$$

**Semantics similarity and syntactic correctness losses** To maintain the semantics of the generated paraphrase similar with the original sample, we adopt a word-level similarity loss based on cosine similarities for the word embeddings between the generated word and the original word.

$$\begin{aligned} \ell_{sem} &= 1 - sim(\mathbf{x}, \mathbf{x}') \\ sim(\mathbf{x}, \mathbf{x}') &= \frac{1}{|\mathbf{x}|} \sum_{i=1}^{|x|} (1 - cos(\mathbf{x}_i, \mathbf{x}'_i)) \end{aligned} \tag{11}$$

The word embeddings similarity cannot guarantee a proper syntactical role of the replacement word. Thus, we also adopt the word composable

**Algorithm 1** The Training Process for End-to-End Adversarial Sample Generator

---

**Input:** Training set $D_t$
**Output:** Condition generator $G$, conditioned paraphrasing model $F$

1: *//Pretrain conditioned paraphrasing model*
2: Generate paraphrasing training data $D_p$ base on $D_t$
3: Pretrain $F$ with $D_p$ according to Eq.4
4: *//Pretrain adversarial discriminator*
5: Pretrain $D$ with $D_t$
6: *//Train the end-to-end adversarial sample generation model*
7: Freeze the parameters of $D$
8: **for** sample $\mathbf{x}, y$ in $D_t$ **do**
9:     Let $\mathbf{c} = G(\mathbf{x})$
10:     Calculate $\ell_{con}$ according to Eq.8
11:     Let $\mathbf{x}' = F(\mathbf{x}, \mathbf{c})$
12:     Calculate $\ell_{rec}, \ell_{pun}, \ell_{sem}, \ell_{syn}$ according to Eq.9-12
13:     Let $\hat{y} = D(\mathbf{x}')$
14:     Calculate $\ell_{adv}$ according to Eq.7
15:     Calculate $L_{adv}$ according to Eq.13
16:     Update $G, F$ according to $L_{adv}$
17: **end for**
18: **return** Condition generator $G$, conditioned paraphrasing model $F$

---

| Dataset | #Class | #Train | #Test | Ave.Len. |
|---------|--------|--------|-------|----------|
| **IMDB** | 2 | 25000 | 25000 | 270.7 |
| **Quora** | 2 | 363660 | 40404 | 25.33 |
| **SNLI** | 3 | 550152 | 10000 | 22.4 |

Table 1: Dataset statistics

The overall training process of the end-to-end adversarial sample generation is listed in Alg.1.

### 3.6 Data Augmentation

Having the above end-to-end adversarial sample generator, we can apply data augmentation on the training data for the target task. For the given original training set $D_t$, the corresponding generated adversarial samples is denoted by $D_t'$. Then the model for the target task can be trained on $D_t \bigcup D_t'$ for improving the model robustness. Since the adversarial samples are more likely to introduce noises, their training loss is assigned a lower weight. Let $L_t$ denotes the loss of the target task, $\lambda_i$ denotes the loss weight for sample $s_i$. $\lambda_i = 1$ for an original sample while $\lambda_i < 1$ for a generated sample.

$$L_{task} = \sum_{s_i \in D_t \bigcup D_t'} \lambda_i L_t(s_i) \qquad (14)$$

The previous work has shown the adversarial samples generated based on one model are potentially adversarial to other models (Szegedy et al., 2013).

## 4 Experiments

To evaluate the effectiveness of the proposed method, four aspects should be considered and evaluated in the subsections 4.2-4.5.

**A1:** To check the attacking effectiveness of generated adversarial sample.

**A2:** To check whether the model performance is improved after augmented training.

**A3:** To check whether the robustness of the target model is improved.

**A4:** To check the efficiency of our method.

### 4.1 Experiment Setups

#### 4.1.1 Tasks

Three tasks are considered as evaluation tasks to cover a wider range of NLP tasks such that the effectiveness of data augmentation can be thoroughly evaluated in different scenarios. The first is a sentiment analysis task on the dataset IMDB. The IMDB

knowledge based loss (Liu et al., 2023) to provide syntactical correctness guidance for the generated sample. Given a dependency relationship between two words $< u, r, v >$, the word composable knowledge estimates the syntactical soundness of this combination, denoted by $p_s(u, r, v)$. It relies on the pretrained word embeddings and thus is easy to integrate with other neural network structures. Let $s_x = (i, r, j)$ denote the dependency parsing tree of $\mathbf{x}$, where $(i, r, j)$ represents the dependency relation between $\mathbf{x}_j$ and its head word $\mathbf{x}_i$ is $r$. The syntactic correctness loss is calculated as follows:

$$\ell_{syn} = - \sum_{(i,r,j) \in s_x} \log p_s(\mathbf{x}_i', r, \mathbf{x}_j') \qquad (12)$$

The complete loss function for training the adaptive adversarial sample generation model is the weighted sum of the above losses.

$$\begin{aligned} L_{adv} = & w_a \ell_{adv} + w_c \ell_{con} + w_r \ell_{rec} + w_p \ell_{pun} \\ & + w_{sem} \ell_{sem} + w_{syn} \ell_{syn} \end{aligned}$$
$$(13)$$

| Base Model | BERT | | | DistilBERT | | | Perturbed % | Bert Score |
|---|---|---|---|---|---|---|---|---|
| Dataset | Ori.Acc. | Att.Acc. | $\Delta$ | Ori.Acc. | Att.Acc. | $\Delta$ | | |
| IMDB | 93.9% | 66.58% | -27.32% | 92.96% | 61.76% | -31.2% | 9.95% | 95.98 |
| Quora | 84.84% | 75.95% | -8.89% | 84.32% | 76.44% | -7.88% | 16.6% | 95.96 |
| SNLI | 88.95% | 55.64% | -33.31% | 88.04% | 55.20% | -32.84% | 14.73% | 95.62 |

Table 2: Accuracy comparison between accuracies on the original testing set and adversarial testing set.

dataset consists of long movie reviews which are classified as positive of negative sentiment polarity. The second is a text classification task on the dataset Quora Question Pairs(QQP), which consists of question pairs from the Quora site classified as duplicate or not. The last is a natural language inference task on the dataset SNLI. Each sample in SNLI consists a hypothesis sentence and a premise sentence, the logical relationship between them is determined as entailment, neutral or contradiction. The statistics of the datasets are listed in Table.1.

### 4.1.2 Base Models

Recently, pretrained language models are widely used as a fundamental part for NLP methods. In the following experiments, BERT (Kenton and Toutanova, 2019) and DistilBERT (Sanh et al., 2019) are chosen as the base model for the tasks. BERT is a widely used pretrained language model, and achieves improvements on various NLP tasks. DistilBERT is a compressed version of BERT model based on model distilling technique, making it easier to be used on devices with lower computation power. Both models will be finetuned on the original train set or the augmented train set for the target task and evaluated afterwards. Their results will be listed as **Baseline** in the following experiments.

### 4.1.3 Implementation

All the experiments are conducted on a single NVIDIA Geforce RTX 2080Ti. Our method is implemented with PyTorch 1.12.0. The code for our proposed method is shared at `https://splab.sdu.edu.cn/xscg/sjjydm.htm`. The compared methods and adversarial sample generation are implemented by TextAttack (Morris et al., 2020b). The size of hidden vectors in our model is 300. We adopt GloVe (Pennington et al., 2014) as the word embeddings and reused the pretrained parameters of word composable knowledge released in SynPara (Liu et al., 2023). For pretraining the paraphrasing model, the ratio of replaced words is set

| Base Model | Method | Dataset | | |
|---|---|---|---|---|
| | | IMDB | Quora | SNLI |
| BERT | **Baseline** | 93.9 | 84.84 | 88.95 |
| | SSMBA | -0.41 | +3.03 | -2.88 |
| | A2T | -0.03 | +1.4 | -0.5 |
| | $A^3$ | **+1** | **+4** | **+0.1** |
| Distil-BERT | **Baseline** | 91.87 | 84.33 | 88.04 |
| | SSMBA | +1.09 | +3.79 | -0.12 |
| | A2T | +1.27 | +4.39 | **+1.38** |
| | $A^3$ | **+1.93** | **+4.89** | +0.22 |

Table 3: Accuracy comparison on the original testing set. The values for the augmentation methods are the differences with the accuracy of the base model.

15% when generating training data. For training the complete model, the weights in the final loss function are set as $w_a = 5.0$, $w_c = 3.0$, $w_r = 2.0$, $w_p = 5.0$, $w_{sem} = 2.0$, $w_{syn} = 0.1$ in most cases. The ratio of perturbed words $\rho$ for the condition generator is set as $0.15$ for SNLI and $0.1$ for IMDB and Quora unless stated otherwise.

### 4.1.4 Compared methods

For the comparison methods, we consider the following methods. **SSMBA** (Ng et al., 2020) is a data augmentation method that uses the corruption function similar to the masked language model training and reconstructs the samples with BERT. **A2T** (Yoo and Qi, 2021) is an adversarial training method, where a fast adversarial sample generation method uses the gradients for identifying important words in a sample and replaces them based on word embeddings or MLM models. It iteratively trains the task model and augment the training data set with the generated adversarial samples.

### 4.2 Attack Effectiveness

To evaluate the effectiveness of the generated adversarial samples, we compared the performances of the baseline models on the original testing data and on the adversarial data. The results are shown in Table.2, where the ratio of perturbed words in

| Base Model | | BERT | | | | | | DistilBERT | | | | | |
| Datasets | | IMDB | | Quora | | SNLI | | IMDB | | Quora | | SNLI | |
| Attack Method | Augment Method | % | Δ | % | Δ | % | Δ | % | Δ | % | Δ | % | Δ |
|---|---|---|---|---|---|---|---|---|---|---|---|---|---|
| $A^3$ | Baseline | 69.3 | | 77.3 | | 58.3 | | 64.1 | | 80.1 | | 59.8 | |
| | SSMBA | 71.6 | +2.3 | 77.7 | +0.4 | 58.7 | +0.4 | 61.6 | -2.5 | 78.8 | -1.3 | 62.9 | +3.1 |
| | A2T | 73.7 | +4.4 | 76.9 | -0.4 | 59.7 | +1.4 | 66.1 | +2 | 78.6 | -1.5 | 60.3 | +0.5 |
| | $A^3$ | **92.2** | **+22.9** | **89.3** | **+12** | **86.2** | **+27.9** | **93.8** | **+29.7** | **90.9** | **+10.8** | **86.5** | **+26.7** |
| A2T | Baseline | 38.8 | | 40.1 | | 44.1 | | 30.6 | | 50.3 | | 41.1 | |
| | SSMBA | 38.9 | +1 | 48.1 | +8 | 49.3 | +5.2 | 27.9 | -2.7 | 50.7 | +0.4 | 45.1 | +4 |
| | A2T | 44 | +5.2 | **76.2** | **+36.1** | **79.4** | **+35.3** | **54.5** | **+23.9** | **76.1** | **+25.8** | **75.6** | **+34.5** |
| | $A^3$ | **45.3** | **+6.5** | 60.7 | +20.6 | 48 | +3.9 | 36.1 | +5.5 | 60.2 | +9.9 | 43.3 | +2.2 |
| Text-Fooler | Baseline | 1.7 | | 23.6 | | 0.9 | | 0.1 | | 10.3 | | 0.47 | |
| | SSMBA | 5.3 | +3.6 | 12.5 | -11.1 | 0.4 | -0.5 | 0 | -0.1 | 12.1 | +1.8 | 0.73 | +0.26 |
| | A2T | 0.8 | -0.9 | 14.3 | -9.3 | **5.1** | **+4.2** | 1.3 | +1.2 | 8 | -2.3 | **3.93** | **+3.46** |
| | $A^3$ | **12** | **+10.3** | **27.1** | **+3.5** | 1.9 | +1 | **1.6** | **+1.5** | **15.2** | **+4.9** | 0.67 | +0.2 |
| BAE | Baseline | 31 | | 39.3 | | 22.1 | | 24.9 | | 39.3 | | 23.87 | |
| | SSMBA | 30.6 | -0.4 | 39 | -0.3 | 25 | +2.9 | 21 | -3.9 | 39.7 | +0.4 | 23.93 | +0.06 |
| | A2T | 29.1 | -1.9 | **43.4** | **+4.1** | 23.5 | +1.4 | 26.3 | +1.4 | 44.2 | +4.9 | **25.5** | **+1.63** |
| | $A^3$ | **37.9** | **+6.9** | 40.8 | +1.5 | **27.6** | **+5.5** | **27.7** | **+2.8** | **45.7** | **+6.4** | 24.27 | +0.4 |
| PWWS | Baseline | 1 | | 25.1 | | 1.2 | | 0.8 | | 22.1 | | 1.07 | |
| | SSMBA | 2.2 | +1.2 | 23.1 | -2 | 2.3 | +1.1 | 0.1 | -0.7 | 22.9 | +0.8 | 1.2 | +0.13 |
| | A2T | 0.3 | -0.7 | 26.3 | +1.2 | 0.7 | -0.5 | 0.6 | -0.2 | 23 | +0.9 | 0.87 | -0.2 |
| | $A^3$ | **7.3** | **+6.3** | **27.7** | **+2.6** | **2.4** | **+1.2** | **0.9** | **+0.1** | **24.1** | **+2** | **2.4** | **+1.33** |

Table 4: Comparison of attacking methods against augmentation methods. Column % denotes the model accuracy under attacking. Column Δ denotes the difference between augmented model with the original model.

the adversarial samples is listed in column *Perturbed%*. A lower ratio indicates more structure information of the text preserved. BertScore (Zhang et al., 2020) is used for quantifying the semantics similarity between the original sample and the adversarial sample. The higher the BertScore, the closer their semantics. We can see that the model performances of both baseline models drop significantly on the adversarial samples, which show that the generated adversarial samples are effective on attacking the baseline models.

### 4.3 Performance Improvement

We then evaluated the model performance on the target tasks before and after training with augmented data. By using our method and SSMBA, the data is augmented to doubled size of the original training set, respectively. The augmented dataset is used for training the base models. For A2T, it iteratively trains the model and then augments the training set by generating adversarial samples. The results in Table.3 show that our method improves the performance on all settings and are better than the compared methods in most cases. The compared methods are unstable on performances, namely in some cases performances are the improved while some are not. For example, SSMBA has negative impact on SNLI dataset for both base models. This is probably because SSMBA doesn't take into con-

sideration the task-specific characteristics, such as the logical connections between the hypothesis and premise in SNLI. It should be also noted that while the original performance of DistilBERT is slightly lower than BERT, it improves a lot after data augmentation, which confirmed previous studies regarding the effectiveness of data augmentation for distilled models (Das et al., 2020).

### 4.4 Robustness Improvement

To evaluate the robustness of base model against different attacking methods, we choose three representative attacking methods: TextFooler (Jin et al., 2020), BAE (Garg and Ramakrishnan, 2020) and PWWS (Ren et al., 2019). We adopt the notion $m1-m2$ denoting the attacking method $m1$ against augmentation method $m2$. For example, BAE-A2T means using BAE to attack a model trained using A2T. Similar to the settings of previous works (Yoo and Qi, 2021), 1000 samples are selected from the original testing set of each dataset (1500 samples from SNLI for three labels), where the label distributions are balanced. To better show the impact of an attacking method, the selected samples can be predicted correctly by the base models before data augmentation. The attack methods generate the adversarial samples based on the selected samples and the accuracy on these adversarial samples of the trained model would be compared. A2T and

| Methods | Dataset | | |
|---|---|---|---|
| | IMDB | Quora | SNLI |
| SSMBA | 10m | 38m | 1h2m |
| A2T | 36h(3) | 111h(2) | 70h(2) |
| $A^3$ | **4m** | **27m** | **21m** |

Table 5: Time for augmenting the dataset. It should be noted that the time for A2T is the cumulated time for multiple training epochs (number of epochs is listed in the brackets).

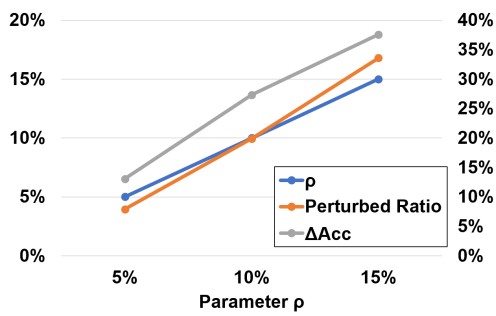

Figure 2: The influence of $\rho$ on the perturbed ratio and the attack effectiveness (denoted by $\Delta$Acc)

our proposed method are tested as they are also adversarial sample generation methods.

The results are listed in Table.4, where the column % denotes the model accuracy under attacking and column $\Delta$ shows the difference between the performance of the augmented model with the baseline. As explored in a previous research (Carlini et al., 2019), the robustness against a certain attack method will be improved if trained with the corresponding adversarial samples. From the row $A^3$-$A^3$ and A2T-A2T, we can see that both methods significantly improve the model robustness. For other representative attacking methods, the augmented models with our generated data show better robustness than the compared methods in most settings.

### 4.5 Efficiency Comparison

Another vital aspect of adversarial training is the efficiency. Most adversarial sample generation methods involves constantly interaction with the target model. Such process is both computational heavy and time-consuming. We recorded the time used for augmenting the training set of different methods. Both SSMBA and our proposed method are one-time augmentation thus the time taken is similar. Our method is slightly faster than SSMBA since SSMBA uses BERT as the reconstruction model. The A2T method, on the other hand, is significantly slower than our method due to two possible rea-

son. First reason is that the attacking process is conducted multiple times. Secondly, the method is based on attacking the target model which is not only time-consuming, but the time taken will also increase over time since the model is trained better after each epoch. Comparably, our proposed method generates model-agnostic adversarial samples, such that the time for augmented training is significantly shortened.

### 4.6 Parameter Analysis

We analyze the effect of parameter $\rho$, the targeted perturbed ratio. In our method for a $\rho$ setting, the actual perturbed ratio in the generated adversarial samples may be different with the initial value. Thus, we analyze the actual perturbed ratio and its impact on the attacking effectiveness, shown in Fig.2. The blue and orange lines represent the set $\rho$ and the actual perturbed ratio respectively, corresponding to the left Y-axis. The $\Delta$Acc is the accuracy difference after the attack, corresponding to the right Y-axis. Higher value of $\Delta$Acc means more effective attack. We can see that the actual perturbed ratio approaches the $\rho$ setting. A higher $\rho$ may cause the predicated label flipped but may lead to a larger semantics difference. Thus, the parameter $\rho$ should be set properly to achieve a balanced performance.

## 5 Conclusion

In this paper, we present a method named $A^3$ that combines the advantages of adversarial training and data augmentation to enhance the robustness of NLP models. We introduce an end-to-end adversarial sample generation model which consists of a condition generator and a conditioned paraphrasing model. A pretrained adversarial discriminator is proposed to train the adversarial sample generation model for capturing the task-specific data characteristics. This method is much efficient without the need for extensive model interactions. The generated adversarial samples are incorporated with the original training samples by a weighted loss for augmented training downstream models. Experiments are conducted on multiple NLP tasks and demonstrate the effectiveness and efficiency of our proposed method. Compared to existing methods, the performances of the base models have improved more after our data augmentation, as well as their robustness enhanced against multiple attacking techniques.

## Limitations

Firstly, the current form of condition limits the diversity and fine-grained control over the generation process. For example, a condition can be a subtree from the dependency syntax structure, which allows paraphrases with potential structure reform and correct syntax (Liu et al., 2023). Secondly, a condition may also contain the direction of replacement word for the paraphrasing model. This would allow reusing the same conditioned paraphrasing model across different tasks, thereby reducing the training overhead for adversarial sample generation. Lastly, in future research, we aim to explore the feasibility of directly using the gradient from the target model for training the adversarial sample generation model.

## Acknowledgments

This work was supported by the National Natural Science Foundation of China (62376138) and the Innovative Development Joint Fund Key Projects of Shandong NSF (ZR2022LZH007).

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
