# OpenReview forum: "End-to-end Adversarial Sample Generation for Data Augmentation"
_EMNLP/2023/Conference — EMNLP 2023 Findings_

### Official Review · Reviewer_CHih · 2023-07-25

**Typos Grammar Style And Presentation Improvements:** 77
**Soundness:** 3

**Excitement:**

2: Mediocre: This paper makes marginal contributions (vs non-contemporaneous work), so I would rather not see it in the conference.

**Missing References:**

Eqn 12: Is the loss normalized based on the length of the syntax tree of x? Will the length of x’ always be the same to x?
Line 379: how do you choose \lambda_i for the generated sample?
Line 441: Why fix the pretrain ratio as 15%? Wouldn’t this affect the fine-tuning stage with different /rho (which is set 0.1 in IMDB and Quora)
Line 443-445: Sensitivity of all six parameters is not evaluated. Ablation study is required.
Line 468: Is perturbed% a metric or statistics of all perturbed words used in adversarial samples? Why is the perturbed ratio in Quora higher but the performance drop is lower?
How do you generate the test set? How do you validate the test set is good enough to present the distribution of adversarial samples?

Is the test set generated using the model after pre-training or after fine-tuning? Do you have confidence intervals for multiple experiments for the results?


Line 558 (Table 5): Does the time calculation consider the pre-training stage? Is the pre-training stage agnostic to datasets/tasks?

**Paper Topic And Main Contributions:**

The author proposes A^3, a framework to generate adversarial data samples. The framework contains a context generator to select which words are going to be replaced, a paraphrasing model to change the words, and a discriminator to justify if the data is adversarial. The above modules require pre-training. The author then uses the framework to perform end-to-end training. In this stage, the author provides six losses for different purposes with almost no ablation studies to the choice of hyperparameter. The author shows that the generated adversarial samples are harmful to the model, and training the model with them will augment the robustness.    However, the efficiency comparison remains questionable, and the parameter analysis of /rho is confusing.

In general, there remains a lot of question unanswered / not well explained in this paper.

**Questions For The Authors:**

Eqn 12: Is the loss normalized based on the length of the syntax tree of x? Will the length of x’ always be the same to x?
Line 379: how do you choose \lambda_i for the generated sample?
Line 441: Why fix the pretrain ratio as 15%? Wouldn’t this affect the fine-tuning stage with different /rho (which is set 0.1 in IMDB and Quora)
Line 443-445: Sensitivity of all six parameters is not evaluated. Ablation study is required.
Line 468: Is perturbed% a metric or statistics of all perturbed words used in adversarial samples? Why is the perturbed ratio in Quora higher but the performance drop is lower?
How do you generate the test set? How do you validate the test set is good enough to present the distribution of adversarial samples?

Is the test set generated using the model after pre-training or after fine-tuning? Do you have confidence intervals for multiple experiments for the results?


Line 558 (Table 5): Does the time calculation consider the pre-training stage? Is the pre-training stage agnostic to datasets/tasks?

**Reasons To Accept:**

The results show some improvements. The motivation of adding each loss and module is clear.

**Reasons To Reject:**

TL;DR I appreciate the efforts and observations / merits found by the authors. However, this paper poorly presents the methodology (both details and its key advantage), and it’s hard to validate the conclusion with such little hyperparameter analysis. I would love to see more detailed results, but I could not accept this version to be accepted as an EMNLP paper.

1 There are too many missing details when presenting the methodology: e.g., what will be the effect if I remove one or two losses presented by the authors? Though their motivations are clear, they do not validate the hypothesis clearly.

2 A lot of equations look like placeholders, such as equations (1, 2, 3, 5, 6).

3 Some of the pieces are simply using existing methods, such as  equation (12), the presentation of these methods are also vague (can only be understood after checking the original paper).

4 The pipeline misses a lot of details. For example, how long does it take to pre-train each module? How adding pre-training will benefit the performance? How to schedule the training of the discriminator and the main module? Not mentioning the detail design of the RNN network used.

5 Why do we need to focus on the four aspects? They are just listed there. Also, some of the results presentation does not seem to be thorough and valid. For example, in table 2, the Quora datasets have the highest perturbation ratio, but the downgraded performance is the least among the three. Is it really because the adversarial samples are effective instead of the task variance or dataset variance?  Also, we didn’t see the attack performance of other comparison methods. And how is the test set generated? What is the size of the adversarial test set and why is that a good benchmark?

6 In table 4, it’s actually hard to say which is better, A^3 or A^2T, if you count the number of winners for each row and column.

7 In table 5, is the computation time also considered the pre-training stage? If not, why? Does the pre-training stage can serve as a unified step which is agnostic to the dataset and tasks?

8 I don’t quite understand the point of section 4.6, and its relationship to the effectiveness of A^3. This influence of /rho seems to be really obvious. I would rather be more interested in changing the six hyperparameters mentioned in line 444 and test their effectiveness.

9 The related work section is also not very well-written. I couldn’t understand what is the key difference and key advantage of A^3 compared to the other methods.

**Reproducibility:**

2: Would be hard pressed to reproduce the results. The contribution depends on data that are simply not available outside the author's institution or consortium; not enough details are provided.

**Reviewer Confidence:**

5: Positive that my evaluation is correct. I read the paper very carefully and I am very familiar with related work.

---

> ### Author Rebuttal · Authors · 2023-08-27
>
> Thanks for the comments and feedbacks. Here are our responses regarding the issues and questions in the review.
>
> R1. There are 6 loss functions during the training process of our model, which makes the space of hyper-parameter combinations enormous for thorough exploration. The current setting is based on the cognitive perspective and validated by experimental results.
>
> R2/3. Thanks for pointing out. The representation will be refined in the later version.
>
> R4. About the training time of our model, the pretraining process takes around 1-2 hours for the conditioned paraphrasing model and 1-2 hours for the adversarial discriminator.  In the end-to-end training process, it takes around 3-4 hours for the complete model on the stated hardware specifications in Sec.4.1.3.
> Currently, the BiRNN adopts 2-layer GRU structure with the hidden dimension of 300. More details will be added in later revisions.
>
> R5. The structure of experimental evaluation is similar with the previous works like A2T.
>
> (About the results of a larger perturbed ratio not bringing a larger performance downgrade on the Quora dataset):
> The current results are based on the multiple experiments, which are related to the characteristics of the dataset.
>
> Since the main purpose of our method is for data augmentation, it is not necessary to compare with other attacking methods.
>
> We have presented the details on how to generate the adversarial test set in paper Lines 514-517, which are similar as the previous works.
> The test set is generated with samples that can be correctly predicted by the fine-tuned base model on the target tasks. Such that the attacked accuracy isn’t affected by the model’s original performance.
>
> R6. Our method outperforms A2T on the A2T-A^3 setting against A2T attacking method and on most settings against BAE/PWWS/Textfooler attacking techniques.
> Meanwhile, A2T appears better than ours mostly on the cases by itself as the attacking technique.
>
> R7. The pretraining stage is not considered since A2T does not have one.
> If the computation time for pre-training stage is taken into account, our method still outperforms baselines, as stated in the above point R4.
>
> R8. Please refer to the above point R1.
>
> R9. Thanks for the suggestion. The representation will be further refined.
>
> Q.1: Yes, the loss is normalized based on the length. The length of x’ is the same as x since the paraphrasing is on word-level.
>
> Q.2: The choice of lambda is based on empirical experiments, 0.2-0.6 for different datasets.
>
> Q.3: The \rho is set 0.15 to balance the preservation of original semantics and variance of expression. We found the \rho setting in the pre-training process doesn’t affect fine-tuning results since the paraphrasing model is also trained during the end-to-end training.
>
> Q.4: Please refer to the above point R1.
>
> Q.5: The perturbed ratio metric refers to the average ratio on all samples, not all perturbed words against all samples.
>
> Q.6-10: Please refer to the above points R5 and R7.

---

### Official Review · Reviewer_ue2L · 2023-08-02

**Soundness:** 3

**Excitement:**

4: Strong: This paper deepens the understanding of some phenomenon or lowers the barriers to an existing research direction.

**Missing References:**

Cite GLoVe on line 437

**Paper Topic And Main Contributions:**

This paper presents an adversarial data generation model for training data augmentation that does not require interaction with the target model.  The authors introduce a three stage model consisting of a BiRNN condition generator, a BiRNN conditioned paraphrasing model, and an adversarial discriminator.   The model is trained end-to-end with a weighted composite loss function to produce adversarial examples that are syntactically correct.  The resulting model can then be used for data augmentation to train a downstream model.

The authors describe the architecture and loss functions in detail, then perform a series of experiments to test the following assumptions: 1) the attacking effectiveness, 2) whether the target model performance is improved after augmented training, 3) whether the robustness of the target model is improved, and 4) the efficiency compared to prior art.  They perform various experiments on three datasets (IMDB, Quora, SNLI) to test the assumptions.  They conclude their methods are superior in quality and efficiency over four comparison adversarial methods and two comparison data augmentation methods.

**Questions For The Authors:**

A: Why did TextFooler and PWWS results perform so much worse in Table 4?

B: In Table 3, DistilBERT seems to enjoy a performance increase regardless of the method used in contrast to BERT.  Is this consistent with prior literature on data augmentation with DistilBERT?

**Reasons To Accept:**

Outside of some rough grammar in places I found this paper to be very easy to follow and the motivations to create the end-to-end system well founded.

The adversarial + augmentation approach appears to greatly outperform the comparison methods in Table 4, of which only one other was also an adversarial + augmentation method (and it performed significantly slower).

The efficiency of the proposed method was significantly improved over the comparison methods, allowing for time and cost reduction over prior approaches in generating large datasets.

**Reasons To Reject:**

In the reported results tables there is no mention if the data is from a single run or best of many.  There is no given standard deviation or CI to frame the variation of quality in the generation between the compared methods.

Reproducibility would be hard if the authors will not release code, incomplete specifics on the BiRNNs used were given in the paper (were the RNNs identical for all stages? optimizer? learning rate? etc).  If no code is to be released, an appendix would be welcome with further details on complete model construction.

**Reproducibility:**

3: Could reproduce the results with some difficulty. The settings of parameters are underspecified or subjectively determined; the training/evaluation data are not widely available.

**Reviewer Confidence:**

4: Quite sure. I tried to check the important points carefully. It's unlikely, though conceivable, that I missed something that should affect my ratings.

**Typos Grammar Style And Presentation Improvements:**

There are many minor grammar issues that can affect clarity, particularly in sections 1, 2.1, 4.5, 4.6, 5.  I would highly recommend running the source through a grammar checker.

It is explained in the text body, but it would help if the y axes were labeled on figure 2.

---

> ### Author Rebuttal · Authors · 2023-08-27
>
> Thanks for the comments and feedbacks. Here are our responses regarding the issues and questions in the review.
>
> R1. The reported results come from the average results of multiple runs.
>
> R2. The source code and datasets will be openly available on the web.
> Currently the BiRNN adopts 2-layer GRU structure with the hidden dimension of 300. More details will be added in later revisions.
>
> Q.A: Both TextFooler and PWWS models replace words in an aggressive try-check way. We generate the adversarial samples in a contextual style since the contextual word replacement is more human-understandable in adversarial sample generation.
>
> Q.B: (About the distilled models benefit more from data augmentation techniques than the original models).
> Yes, it has been studied in previous work:
> Das, Deepan, et al. "An empirical analysis of the impact of data augmentation on knowledge distillation." arXiv preprint arXiv:2006.03810 (2020).

---

### Official Review · Reviewer_m7F9 · 2023-08-05

**Soundness:** 3

**Excitement:**

3: Ambivalent: It has merits (e.g., it reports state-of-the-art results, the idea is nice), but there are key weaknesses (e.g., it describes incremental work), and it can significantly benefit from another round of revision. However, I won't object to accepting it if my co-reviewers champion it.

**Missing References:**

- Lantao Yu, Weinan Zhang, Jun Wang, and Yong Yu. 2017. Seqgan: Sequence generative adversarial nets with policy gradient. In Proceedings of the AAAI conference on artificial intelligence, volume 31.
- Danilo Croce, Giuseppe Castellucci, and Roberto Basili. 2020. GAN-BERT: Generative adversarial learning for robust text classification with a bunch of labeled examples. In Proceedings of the 58th Annual Meeting of the Association for Computational Linguistics, pages 2114–2119, Online. Association for Computational Linguistics.
- BERTScore: https://arxiv.org/abs/1904.09675

**Paper Topic And Main Contributions:**

The paper proposes a novel approach based on adversarial training and data augmentation. This approach consists of conditioned paraphrasing model and condition generator to generate adversarial sample. Pre-trained discriminator is applied together to adapt to the different downstream tasks. The goal of this approach is to generate the robust model against to the diverse adversarial attacks.

**Questions For The Authors:**

A. Can you clarify the discriminator in your approach? Is it based on pre-trained model or trained from scratch?
B. Why did you consider the bidirectional RNN, instead of transformer in condition generator and conditioned paraphrasing model?
C. Why did you only consider paraphrasing model for adversarial generation? How about other text generation methods like generating the Out-of-domain words?
D. In my perspective, paraphrasing is affected by sentence-level, not only word-level. Why did you generate the condition or rephrased sentence based on word? For word-level, I guess rule-based approach is better one.
E. In line 485, how does the size of train set in A2T?
F. Why A2T in SNLI+DistillBERT is better than your approach in Table 3?

**Reasons To Accept:**

- Generate the conditions for paraphrasing model to make a suitable adversarial sample.
- Test on three different tasks to reveal the novelty of works.
- Simple but powerful performances regarding the experiments done by authors.

**Reasons To Reject:**

- Idea is very similar to Generative Adversarial Network. The author needs to point out the difference to show the novelty of work in the Related work sections better. Please see "Missing References" part.
- The algorithm is not clear, especially for discriminator since author sometimes argues that it is pre-trained model (line 82-85) but sometimes not (line 5 in Algorithm 1). Please develop the algorithm part better to make it more understandable.
- The insights of authors choices (e.g. model selection) and the investigation of experimental results are missing. For example, why A2T in SNLI+DistillBERT is better than your approach (Table 3)?
- Please include the example of adversarial examples in each approach to understand better.
- Lack of ablation study regarding the selection of hyperparameters (e.g. Section 4.1.3).
- Hard to reproduce the results since there is no offered source code.
- Please double-check whether authors follow the ACL format since the labelling of Table/Figure/Algorithm does not work.

**Reproducibility:**

2: Would be hard pressed to reproduce the results. The contribution depends on data that are simply not available outside the author's institution or consortium; not enough details are provided.

**Reviewer Confidence:**

4: Quite sure. I tried to check the important points carefully. It's unlikely, though conceivable, that I missed something that should affect my ratings.

**Typos Grammar Style And Presentation Improvements:**

- In Figure 1, "dot (.)" is missing.
- Add space in data augmentationA^3 (line 77)
- Authors sometimes use pretrained (line 82) but sometimes use pre-trained (line 157). Please use it consistently.

---

> ### Author Rebuttal · Authors · 2023-08-27
>
> Thanks for the comments and feedback. Here are our responses regarding the issues and questions in the reviews.
>
> R1. Firstly, there is the distinct difference with GAN. The goal of GAN discriminators is to distinguish samples being real or fake, or belonging to different categories. Differently, the discriminator in our work is task-dependent and the purpose is making the task model predict incorrectly.
> Secondly, comparing with the mostly related works like GAN-BERT, our model generates the fine-grained controllable and understandable sentences, namely samples, which they do not.
>
> R2. The term of “pre-training” in our method refers to process of training the discriminator on the target task before the complete model training.
>
> R3. (The reason about A2T having better performance on the specific SNLI task)
> SNLI has higher requirements on the logical relationships between the hypothesis and premise. A2T constantly interact with the target model during the generation of the adversarial sample. But it is time consuming and not transferable across different models compared to our method.
>
> R4. The generated samples are not included in this version due to the space limitation. We would present some illustrative samples in the appendices later.
>
> R5. About the ablation experiments on the hyper-parameter settings, since there are 6 loss functions, the space of their combination is enormous for exploration and thorough validation.
> Currently, we try different combinations and set them based on both the experimental results and the cognitive perspectives.
>
> R6. The source code and datasets will be openly available on the web.
>
> R7. About the problem of labeling and jumping for table/figure/algorithm in the downloaded pdf version of our paper, they work well on our side. We use Acrobat DC.
>
> Q.A: Please refer to the above point R2.
>
> Q.B: The BiRNN is selected as the model due to its simplicity and cost-effectiveness with a small number of parameters. Since the target of our method is to provide task-dependent data augmentation, in practice there is not much labelled data. Thus, it requires the model to be light-weight for training rather than the transformer based large models.
>
> Q.C/D: Previous works in data augmentation and adversarial sample generation prove the effectiveness of word substitution-based paraphrasing. It’s easier to maintain the semantic similarity and reduce the complexity of generating plausible samples than sentence level generation. In this paper, we combine the word-level paraphrasing with the complete sentence-level generation.
> Considering the rule-based approaches, they cannot be trained in an end-to-end style like our proposed method.
>
> Q.E: A2T does not require a training process.
>
> Q.F: Please refer to the above point R3.

---

### Meta-Review · Area_Chair_ToCJ · 2023-09-19

**Recommendation:** 3

**Metareview:**

This paper proposed a GAN-like approach to generate adversarial samples. These augmented samples are used during training to enhance robustness of the target model in front of various attacking techniques. The reviewers find this work overall sound but with some issues on details, clarify, and ablations, as summarized below.

### Pros
1. "Simple but powerful performances regarding the experiments done by authors."
2. "The efficiency of the proposed method was significantly improved over the comparison methods, allowing for time and cost reduction over prior approaches in generating large datasets."
3. "Test on three different tasks to reveal the novelty of works."


### Cons
1. There are many clarity issues such as missing details, missing ablations and writing problems raised by Reviewer m7F9 and Reviewer CHih, as reflected by relatively low reproducibility scores as well.

2. Ablation analysis is not comprehensive enough to justify authors' choices.

Overall, I think this paper sound but not exciting enough.

---

### Decision · Program_Chairs · 2023-10-07

**Decision:**

Accept-Findings

**Comment:**

This paper proposed a GAN-like approach to generate adversarial samples. These augmented samples are used during training to enhance robustness of the target model in front of various attacking techniques. The reviewers find this work overall sound but with some issues on details, clarify, and ablations, as summarized below.

### Pros
1. "Simple but powerful performances regarding the experiments done by authors."
2. "The efficiency of the proposed method was significantly improved over the comparison methods, allowing for time and cost reduction over prior approaches in generating large datasets."
3. "Test on three different tasks to reveal the novelty of works."


### Cons
1. There are many clarity issues such as missing details, missing ablations and writing problems raised by Reviewer m7F9 and Reviewer CHih, as reflected by relatively low reproducibility scores as well.

2. Ablation analysis is not comprehensive enough to justify authors' choices.

Overall, I think this paper sound but not exciting enough.